# Differentiable Clustering with Perturbed Spanning Forests

**Lawrence Stewart**[*]
ENS & INRIA
Paris, France

**Francis Bach**
ENS & INRIA
Paris, France

**Felipe Llinares-López**
Google DeepMind
Paris, France

**Quentin Berthet**
Google DeepMind
Paris, France

## Abstract

We introduce a differentiable clustering method based on stochastic perturbations of minimum-weight spanning forests. This allows us to include clustering in end-to-end trainable pipelines, with efficient gradients. We show that our method performs well even in difficult settings, such as data sets with high noise and challenging geometries. We also formulate an ad hoc loss to efficiently learn from partial clustering data using this operation. We demonstrate its performance on several data sets for supervised and semi-supervised tasks.[2].

## 1 Introduction

Clustering is one of the most classical tasks in data processing, and one of the fundamental methods in unsupervised learning (Hastie et al., 2009). In most formulations, the problem consists in partitioning a collection of $n$ elements into $k$ clusters, in a manner that optimizes some criterion, such as intra-cluster proximity, or some resemblance criterion, such as a pairwise similarity matrix. This procedure is naturally related to other tasks in machine learning, either by using these induced classes in supervised problems, or by either evaluating or looking for well-clustered representations (Caron et al., 2018; Xie et al., 2016). Its performance and flexibility on a wide range of natural dataset, that makes it a good downstream or preprocessing task, also make it a a very important candidate to learn representations in a supervised fashion. Yet, it is a fundamentally "combinatorial" problem, representing a discrete decision, much like many classical algorithmic methods (e.g., sorting, taking nearest neighbours, dynamic programming).

For these reasons, it is extremely challenging to *learn through clustering*. As a function, the solution of a clustering problem is piece-wise constant with respect to its inputs (such as a similarity or distance matrix), and its gradient would therefore be zero almost everywhere. This operation is therefore naturally ill-suited to the use of gradient-based approaches to minimize an objective, which are at the center of optimization procedures for machine learning. It does not have the convenient properties of commonly used operations in end-to-end differentiable systems, such as smoothness and differentiability. Another challenge of using clustering as part of a learning pipeline is perhaps its *ambiguity*: even for a given notion of distance or similarity between elements, there are several valid definitions of clustering, with criteria adapted to different uses. Popular methods include $k$-means, whose formulation is NP-hard in general even in simple settings (Drineas et al., 2004), and relies on heuristics that depend heavily on their initialization (Arthur and Vassilvitskii, 2007; Bubeck et al., 2012). Several of them rely on proximity to a *centroid*, or *prototype*, i.e., a vector representing each cluster. These fail on challenging geometries, e.g., interlaced clusters with no linear separation.

We propose a new method for differentiable clustering that efficiently addresses these difficulties. It is a principled, deterministic operation: it is based on minimum-weight spanning forests, a variant

---

[*]All correspondence should be addressed to lawrence.stewart@ens.fr.

[2]Code base: `https://github.com/LawrenceMMStewart/DiffClust_NeurIPS2023`

37th Conference on Neural Information Processing Systems (NeurIPS 2023).

of minimum spanning trees. We chose this primitive at the heart of our method because it can be represented as a linear program (LP), and this is particularly well-adapted to our smoothing technique. However, we use a greedy algorithm to solve it exactly and efficiently (rather than solving it as an LP or relying on an uncertain heuristic). We observe that this method, often referred to as single linkage clustering (Gower and Ross, 1969), is effective as a clustering method in several challenging settings. Further, we are able to create a differentiable version of this operation, by introducing stochastic perturbations on the similarity matrix (the cost of the LP). This proxy has several convenient properties: it approximates the original function, it is smooth (both of these are controlled by a temperature parameter), and both the perturbed optimizers and its derivatives can be efficiently estimated with Monte-Carlo estimators (see, e.g., Hazan et al., 2016; Berthet et al., 2020, and references therein). This allows us to include this operation in end-to-end differentiable machine learning pipelines, and we show that this method is both efficient and performs well at capturing the clustered aspect of natural data, in several tasks.

Our work is part of an effort to include unconventional operations in model training loops based on gradient computations. These include discrete operators such as optimal transport, dynamic time-warping and other dynamic programs (Cuturi, 2013; Cuturi and Blondel, 2017; Mensch and Blondel, 2018; Blondel et al., 2020b; Vlastelica et al., 2019; Paulus et al., 2020; Sander et al., 2023; Shvetsova et al., 2023), to ease their inclusion in end-to-end differentiable pipelines that can be trained with first-order methods in fields such as computer vision, audio processing, biology, and physical simulators (Cordonnier et al., 2021; Kumar et al., 2021; Carr et al., 2021; Le Lidec et al., 2021; Baid et al., 2023; Llinares-López et al., 2023) and other optimization algorithms (Dubois-Taine et al., 2022). Other related methods, including some based on convex relaxations and on optimal transport, aim to minimize an objective involving a neural network in order to perform clustering, with centroids (Caron et al., 2020; Genevay et al., 2019; Xie et al., 2016). Most of them involve optimization *in order to* cluster, without explicitly optimizing *through* the clustering operation, allowing to learn using some supervision. Another line of research is focused on using the matrix tree-theorem to compute marginals of the Gibbs distribution over spanning trees (Koo et al., 2007; Zmigrod et al., 2021). It is related to our perturbation smoothing technique (which yields a different distribution), and can also be used to learn using tree supervision. The main difference is that it does not address the clustering problem, and that optimizing in this setting involves sophisticated linear algebra computations based on determinants, which is not as efficient and convenient to implement as our method based on sampling and greedy algorithms.

There is an important conceptual difference with the methodology described by Berthet et al. (2020), and other recent works based on it that use *Fenchel-Young losses* (Blondel et al., 2020a): while one of the core discrete operations on which our clustering procedure is based is an LP, the cluster connectivity matrix–which, importantly, has the same structure as any ground-truth label information–is not. This weak supervision is a natural obstacle that we have to overcome: on the one hand, it is reasonable to expect clustering data to come in this form, i.e., only stating whether some elements belong to the same cluster or not, which is weaker than ground-truth spanning forest information would be. On the other hand, linear problems on cluster connectivity matrices, such as MAXCUT, are notoriously NP-hard (Karp, 1972), so we cannot use perturbed linear oracles over these sets (at least exactly). In order to handle these two situations together, we design a *partial Fenchel-Young loss*, inspired by the literature on weak supervision, that allows us to use a loss designed for spanning forest structured prediction, even though we have the less informative cluster connectivity information. In our experiments, we show that our method enables us to learn through a clustering operation, i.e., that we can find representations of the data for which clustering of the data will match with some ground-truth clustering data. We apply this to both supervised settings, including some illustrations on synthetic challenging data, and semi-supervised settings on real data, focusing on settings with a low number of labeled instances and unlabeled classes.

**Main Contributions.** In this work, we introduce an efficient and principled technique for differentiable clustering. In summary, we make the following contributions:

- Our method is based on using spanning forests, and a **differentiable** proxy, obtained by adding **stochastic perturbations** on the edge costs.

- Our method allows us to **learn through clustering**: one can train a model to learn 'clusterable' representations of the data in an **online** fashion. The model generating the representations is informed by gradients that are transmitted through our clustering operation.

- We derive a partial Fenchel-Young loss, which allows us to incorporate **weak cluster information**, and can be used in any weakly supervised structured prediction problem.

- We show that it is a powerful clustering technique, allowing us to learn meaningful clustered representations. It **does not require** a model to learn linearly separable representations.

- Our operation can be incorporated efficiently into any gradient-based ML pipeline. We demonstrate this in the context of **supervised** and **semi-supervised cluster learning tasks**.

**Notations.** For a positive integer $n$, we denote by $[n]$ the set $\{1, \ldots, n\}$. We consider all graphs to be undirected. For the complete graph with $n$ vertices $K_n$ over nodes $[n]$, we denote by $\mathcal{T}$ the set of *spanning trees* on $K_n$, i.e., subgraphs with no cycles and one connected component. For any positive integer $k \leq n$ we also denote by $\mathcal{C}_k$ the set of $k$-*spanning forests* of $K_n$, i.e., subgraphs with no cycles and $k$ connected components. With a slight abuse of notation, we also refer to $\mathcal{T}$ and $\mathcal{C}_k$ for the set of *adjacency matrices* of these graphs. For two nodes $i$ and $j$ in a general graph, we write $i \sim j$ if the two nodes are in the same connected component. We denote by $\mathcal{S}_n$ the set of $n \times n$ symmetric matrices.

## 2 Differentiable clustering

We provide a **differentiable** operator for clustering elements based on a similarity matrix. Our method is **explicit**: it relies on a linear programming primitive, not on a heuristic to solve another problem (e.g., $k$-means). It is **label blind**: our solution is represented as a connectivity matrix, and is not affected by a permutation of the clusters. Finally it is **geometrically flexible**: it is based on single linkage, and does not rely on proximity to a centroid, or linear separability to include several elements in the same cluster (as illustrated in Section 4).

In order to cluster $n$ elements in $k$ clusters, we define a clustering operator as a function $M_k^*(\cdot)$ taking an $n \times n$ symmetric similarity matrix as input (e.g., negative pairwise square distances) and outputting a cluster connectivity matrix of the same size (see Definition 1). We also introduce its differentiable proxy $M_{k,\varepsilon}^*(\cdot)$. We use them as a method to **learn through clustering**. As an example, in the supervised learning setting, we consider the $n$ elements described by features $\mathbf{X} = (X_1, \ldots, X_n)$ in some feature space $\mathcal{X}$, and ground-truth clustering data as an $n \times n$ matrix $M_\Omega$ (with either complete or partial information, see Definition 4). A parameterized model produces a similarity matrix $\Sigma = \Sigma_w(\mathbf{X})$–e.g., based on pairwise square distances between embeddings $\Phi_w(X_i)$ for some model $\Phi_w$. Minimizing in $w$ a loss so that $M_k^*(\Sigma = \Sigma_w(\mathbf{X})) \approx M_\Omega$ allows to train a model based on the clustering information (see Figure 1). We describe in this section and the following one the tools that we introduce to achieve this goal, and show in Section 4 experimental results on real data.

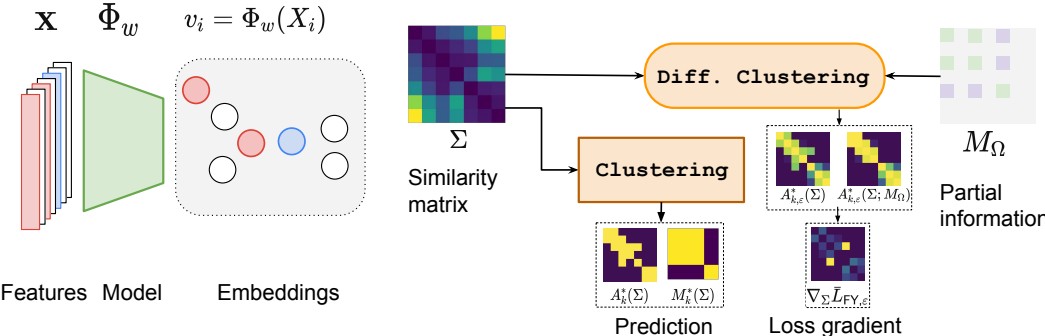

Figure 1: Our pipeline, in the semi-supervised learning setting: data points are embedded by a parameterized model, which produces a similarity matrix. Partial clustering information may be available, in the form of must-link or must-not-link constraints. Our clustering and differentiable clustering operators are used, respectively for prediction and gradient computations.

## 2.1 Clustering with $k$-spanning forests

In this work, the core operation is to cluster $n$ elements, using a similarity matrix $\Sigma \in \mathcal{S}_n$. Informally, pairs of elements $(i, j)$ with high similarity $\Sigma_{ij}$ should be more likely to be in the same cluster than those with low similarity. The clustering is represented in the following manner.

**Definition 1** (Cluster connectivity matrix). *Let $\pi : [n] \to [k]$ be a clustering function, assigning elements to one of $k$ clusters. We represent it with an $n \times n$ binary matrix $M$ (the cluster connectivity):*

$$M_{ij} = 1 \quad \text{if and only if} \quad \pi(i) = \pi(j).$$

*We denote by $\mathcal{B}_k \subseteq \{0, 1\}^{n \times n}$ the set of binary cluster connectivity matrices with $k$ clusters.*

Using this definition, we define an operation $M_k^*(\cdot)$ mapping a similarity matrix $\Sigma$ to a clustering, in the form of such a membership matrix. Up to a permutation (i.e., naming the clusters), $M$ allows to recover $\pi$ entirely. It is based on a maximum spanning forest primitive $A_k^*(\cdot)$, and both are defined below. We recall that a $k$-forest on a graph is a subgraph with no cycles consisting in $k$ connected components, potentially single nodes (see Figure 2). The cluster connectivity matrix is sometimes referred to as the cluster coincidence matrix matrix in other literature, and the two terms can be used interchangeably.

**Definition 2** (Maximum $k$-spanning forest). *Let $\Sigma$ be an $n \times n$ similarity matrix. We denote by $A_k^*(\Sigma)$ the adjacency matrix of the $k$-spanning forest with maximum similarity, defined as*

$$A_k^*(\Sigma) = \underset{A \in \mathcal{C}_k}{\operatorname{argmax}} \langle A, \Sigma \rangle.$$

*This defines a mapping $A_k^* : \mathcal{S}_n \to \mathcal{C}_k$. We denote by $F_k$ the value of this maximum, i.e.,*

$$F_k(\Sigma) = \max_{A \in \mathcal{C}_k} \langle A, \Sigma \rangle.$$

**Definition 3** (Spanning forest clustering). *Let $A$ be the adjacency matrix of a $k$-spanning forest. We denote by $M^*(A)$ the connectivity matrix of the $k$ connected components of $A$, i.e.,*

$$M_{ij} = 1 \quad \text{if and only if} \quad i \sim j.$$

*Given an $n \times n$ similarity matrix $\Sigma$, we denote by $M_k^*(\Sigma) \in \mathcal{B}_k$ the clustering induced by the maximum $k$-spanning forest, defined by*

$$M_k^*(\Sigma) = M^*(A_k^*(\Sigma)).$$

*This defines a mapping $M_k^* : \mathcal{S}_n \to \mathcal{B}_k$, our clustering operator.*

**Remarks.** The solution of the linear program is unique almost everywhere for similarity matrices (relevant for us, as we use stochastic perturbations in learning). Both these operators can be computed using Kruskal's algorithm to find a minimum spanning tree in a graph (Kruskal, 1956). This algorithm builds the tree by iteratively adding edges in a greedy fashion, maintaining non-cyclicity. On a connected graph on $n$ nodes, after $n - 1$ edge additions, a spanning tree (i.e., that covers all nodes) is obtained. The greedy algorithm is optimal for this problem, which can be proved by showing that forests can be seen as the independent sets of a *matroid* (Cormen et al., 2022). Further, stopping the algorithm after $n - k$ edge additions yields a forest consisting of $k$ trees (possibly singletons), together spanning the $n$ nodes and which is optimal for the problem in Definition 2. As in several other clustering methods, we specify in ours the number of desired clusters $k$, but a consequence of our algorithmic choice is that one can compute the clustering operator *for all $k$* without much computational overhead, and that this number can easily be tuned by validation.

Further, a common manner to run this algorithm is to keep track of the constructed connected components, therefore both $A_k^*(\Sigma)$ and $M_k^*(\Sigma)$ are actually obtained by this algorithm. The mapping $M^*$ is of course many-to-one, and yields a partition of $\mathcal{C}_k$ into equivalence classes of $k$-forests that yield the same clusters. This point is actually at the center of our operators of clustering with constraints in Definition 5 and of our designed loss in Section 3.2, both below. We note that as the maximizer of a linear program, when the solution is unique, we have $\nabla_\Sigma F_k(\Sigma) = A_k^*(\Sigma)$.

As described above, our main goal is to enable the inclusion of these operations in end-to-end differentiable pipelines (see Figure 1). We include two important aspects to our framework in order to do so, and to efficiently learn through clustering. The first one is the inclusion of constraints, enforced values coming either from partial or complete clustering information (see below, Definition 5), and the second one is the creation of a differentiable version of these operators, using stochastic perturbations (see Section 2.2).

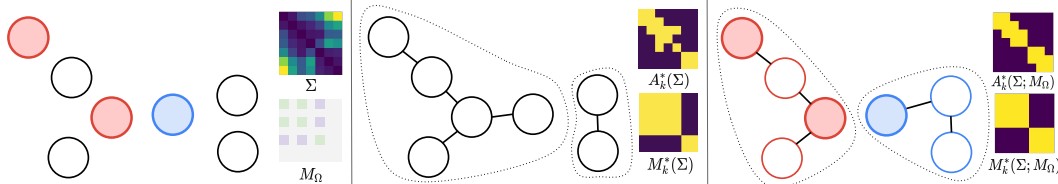

Figure 2: Method illustration, for $k = 2$: **(left)** Similarity matrix based on pairwise square distance, partial cluster connectivity information. **(center)** Clustering using spanning forests without partial clustering constraints. **(right)** Constrained clustering with partial constraints.

**Clustering with constraints.** Given information, we also consider constrained versions of these two problems. We represent the enforced connectivity information as a matrix $M_\Omega$, defined as follows.

**Definition 4** (Partial cluster coincidence matrix). *Let $M$ be a cluster connectivity matrix, and $\Omega$ a subset of $[n] \times [n]$, representing the set of observations. We denote by $M_\Omega$ the $n \times n$ matrix*

$$M_{\Omega,ij} = M_{ij} \quad \text{if } (i,j) \in \Omega, \quad M_{\Omega,ij} = * \quad \text{otherwise} .$$

**Remarks.** The symbol "$*$" in this definition is only used as a placeholder, an indicator of "no information", and does not have a mathematical use. A common setting is where only a subset $S \subseteq [n]$ of the data has label information. In this case for any $i, j \in S$, $M_{\Omega,ij} = 1$ if $i$ and $j$ share the same label, otherwise $M_{\Omega,ij} = 0$ i.e. elements in the same class are clustered together and separated from elements in other classes.

**Definition 5** (Constrained maximum $k$-spanning forest). *Let $\Sigma$ be an $n \times n$ similarity matrix. We denote by $A_k^*(\Sigma\,;M_\Omega)$ the adjacency matrix of the $k$-spanning forest with maximum similarity, constrained to satisfy the connectivity constraints in $M_\Omega$. It is defined as*

$$A_k^*(\Sigma\,;M_\Omega) = \underset{A \in \mathcal{C}_k(M_\Omega)}{\text{argmax}} \langle A, \Sigma \rangle ,$$

*where for any partial clustering matrix $M_\Omega$, $\mathcal{C}_k(M_\Omega)$ is the set of $k$-spanning forests whose clusters agree with $M_\Omega$, i.e.,*

$$\mathcal{C}_k(M_\Omega) = \{A \in \mathcal{C}_k \,:\, M^*(A)_{ij} = M_{\Omega,ij} \,\forall\, (i,j) \in \Omega\} .$$

*For any partial connectivity matrix $M_\Omega$, this defines another mapping $A^*(\,\cdot\,;\,M_\Omega) : \mathcal{S}_n \to \mathcal{C}_k$.*

*We denote by $F_k(\Sigma\,;M_\Omega)$ the value of this maximum, i.e.,*

$$F_k(\Sigma\,;M_\Omega) = \underset{A \in \mathcal{C}_k(M_\Omega)}{\max} \langle A, \Sigma \rangle .$$

*Similarly we define $M_k^*(\Sigma\,;M_\Omega) = M^*(A_k^*(\Sigma\,;M_\Omega))$.*

**Remarks.** We consider these operations in order to infer clusterings and spanning forests that are consistent with observed information. That is particularly important when designing a partial loss to learn from observed clustering information. Note that depending on what the set of observations $\Omega$ is, these constraints can be more or less subtle to satisfy. For certain sets of constraints $\Omega$, when the matroid structure is preserved, we can obtain $A_k^*(\Sigma\,;M_\Omega)$ by the usual greedy algorithm, by additionally checking that any new edge does not violate the constraints defined by $\Omega$. This is for example the case when $\Omega$ corresponds to exactly observing a fully clustered subset of points.

## 2.2 Perturbed clustering

The clustering operations defined above are efficient and perform well (see Figure 2) but by their nature as discrete operators, they have a major drawback: they are piece-wise constant and as such cannot be conveniently included in end-to-end differentiable pipelines, such as those used to train models such as neural networks. To overcome this obstacle, we use a proxy for our operators, by introducing a *perturbed* version (Abernethy et al., 2016; Berthet et al., 2020; Paulus et al., 2020; Struminsky et al., 2021), obtained by taking the expectation of solutions with stochastic additive noise on the input. In these definitions and the following, we consider $Z \sim \mu$ from a distribution with positive, differentiable density over $\mathcal{S}_n$, and $\varepsilon > 0$.

**Definition 6.** *We define the* perturbed maximum spanning forest *as the expected maximum spanning forest under stochastic perturbation on the inputs. Formally, for a similarity matrix $\Sigma$, we have*

$$A_{k,\varepsilon}^*(\Sigma) = \mathbf{E}[A_k^*(\Sigma + \varepsilon Z)] = \mathbf{E}\Big[\operatorname*{argmax}_{A \in \mathcal{C}_k}\langle A, \Sigma + \varepsilon Z\rangle\Big], \quad F_{k,\varepsilon}(\Sigma) = \mathbf{E}[F_k(\Sigma + \varepsilon Z)].$$

*We define analogously $A_{k,\varepsilon}^*(\Sigma\,;M_\Omega) = \mathbf{E}[A_k^*(\Sigma + \varepsilon Z;M_\Omega)]$ and $F_{k,\varepsilon}(\Sigma\,;M_\Omega) = \mathbf{E}[F_k(\Sigma + \varepsilon Z;M_\Omega)]$, as well as clustering $M_{k,\varepsilon}^*(\Sigma) = \mathbf{E}[M_k^*(\Sigma + \varepsilon Z)]$ and $M_{k,\varepsilon}^*(\Sigma\,;M_\Omega) = \mathbf{E}[M_k^*(\Sigma + \varepsilon Z;M_\Omega)]$.*

We note that this defines operations $A_{k,\varepsilon}^*(\cdot)$ and $A_{k,\varepsilon}^*(\,\cdot\,;M_\Omega)$ mapping $\Sigma \in \mathcal{S}_n$ to $\mathrm{cvx}(\mathcal{C}_k)$, the *convex hull* of $\mathcal{C}_k$. These operators have several advantageous features: They are differentiable, and both their values and their derivatives can be estimated using Monte-Carlo methods, by averaging copies of $A_k^*(\Sigma + \varepsilon Z^{(i)})$. These operators are the ones used to compute the gradient of the loss that we design to learn from clustering (see Definition 8 and Proposition 1).

This is particularly convenient, as it does not require to implement a different algorithm to compute the differentiable version. Moreover, the use of parallelization in modern computing hardware makes the computational overhead almost nonexistent. These methods are part of a large literature on using perturbations in optimizers such as LP solutions (Papandreou and Yuille, 2011; Hazan et al., 2013; Gane et al., 2014; Hazan et al., 2016), including so-called Gumbel max-tricks (Gumbel, 1954; Maddison et al., 2016; Jang et al., 2017; Huijben et al., 2022; Blondel et al., 2022)

Since $M_{k,\varepsilon}^*(\cdot)$ is a differentiable operator from $\mathcal{S}_n$ to $\mathrm{cvx}(\mathcal{B}_k)$, it is possible to use any loss function on $\mathrm{cvx}(\mathcal{B}_k)$ to design a loss based on $\Sigma$ and some ground-truth clustering information $M_\Omega$, such as $L(\Sigma\,;M_\Omega) = \|M_{k,\varepsilon}^*(\Sigma) - M_\Omega\|_2^2$. This flexibility is one of the advantages of our method. In the following section, we introduce a loss tailored to be efficient to compute and performant in several learning tasks, that we call a *partial Fenchel-Young loss*.

## 3 Learning with differentiable clustering

### 3.1 Fenchel-Young losses

In structured prediction, a common *modus operandi* is to minimize a loss between some structured *ground truth response* or *label* $y \in \mathcal{Y}$ and a *score* $\theta \in \mathbb{R}^d$ (the latter often itself the output of a parameterized model). As an example, in logistic regression $y \in \{0, 1\}$ and $\theta = \langle x, \beta \rangle \in \mathbb{R}$. The framework of *Fenchel-Young* losses allows to tackle much more complex structures, such as cluster connectivity information.

**Definition 7** (Fenchel-Young loss–Blondel et al. (2020a)). *Let $F$ be a convex function on $\mathbb{R}^d$, and $F^*$ its Fenchel dual on a convex set $\mathcal{C} \subset \mathbb{R}^d$. The Fenchel-Young loss between $\theta \in \mathbb{R}^d$ and $y \in int(\mathcal{C})$ is*

$$L_{\mathsf{FY}}(\theta; y) = F(\theta) - \langle \theta, y \rangle + F^*(y).$$

The Fenchel-Young (FY) loss satisfies several properties, making it useful for learning. In particular, it is nonnegative, convex in $\theta$, handles well noisy labels, and its gradient with respect to the score can be efficiently computed, with $\nabla_\theta L_{\mathsf{FY}}(\theta; y) = \nabla_\theta F(\theta) - y$ (see, e.g., Blondel et al., 2020a).

In the case of linear programs over a polytope $\mathcal{C}$, taking $F$ to be the so-called support function defined as $F(\theta) = \max_{y \in \mathcal{C}}\langle y, \mathcal{C}\rangle$ (consistent with our notation so far), the dual function $F^*$ is the indicator function of $\mathcal{C}$ (Rockafellar, 1997), and the Fenchel-Young function is then given by

$$L_{\mathsf{FY}}(\theta, y) = \begin{cases} F(\theta) - \langle \theta, y \rangle & \text{if } y \in \mathcal{C}, \\ +\infty & \text{otherwise}. \end{cases}$$

In the case of a perturbed maximum for $F$, see, e.g., Berthet et al. (2020), we have

$$F_\varepsilon(\theta) = \mathbf{E}[\max_{y \in \mathcal{C}}\langle y, \theta + \varepsilon Z\rangle], \quad \text{and} \quad y_\varepsilon^*(\theta) = \mathbf{E}[\operatorname*{argmax}_{y \in \mathcal{C}}\langle y, \theta + \varepsilon Z\rangle].$$

In this setting (under mild regularity assumptions on the noise distribution and $\mathcal{C}$), we have that $\varepsilon\Omega = F_\varepsilon^*$ is a strictly convex Legendre function on $\mathcal{C}$ and $y_\varepsilon^*(\theta) = \nabla_\theta F_\varepsilon(\theta)$. This is actually equivalent to having max and argmax with a $-\varepsilon\Omega(y)$ regularization (see Berthet et al., 2020, Propositions 2.1

and 2.2). In this case, the FY loss is given by $L_{\mathsf{FY},\varepsilon}(\theta;y) = F_\varepsilon(\theta) - \langle\theta, y\rangle + \varepsilon\Omega(y)$ and its gradient by $\nabla_\theta L_{\mathsf{FY},\varepsilon}(\theta;y) = y_\varepsilon^*(\theta) - y$. One can also define it as $L_{\mathsf{FY},\varepsilon}(\theta;y) = \mathbf{E}[L_{\mathsf{FY}}(\theta + \varepsilon Z;y)]$ and it has the same gradient. In the perturbations case, it can be easily obtained by Monte-Carlo estimates of the perturbed optimizer, taking $\frac{1}{B}\sum_{i=1}^B y^*(\theta + \varepsilon Z_i) - y$.

## 3.2 Partial Fenchel-Young losses

As noted above, this loss is widely applicable in structured prediction. As presented here, it requires label data $y$ of the same kind than the optimizer $y^*$. In our setting, the linear program is over spanning $k$-forests rather than connectivity matrix. This is no accident: linear programs over cut matrices (when $k = 2$) are already NP-hard (Karp, 1972). If one has access to richer data (such as ground-truth spanning forest information), one can ignore the operator $M^*$ and focus only on $A_{k,\varepsilon}^*$, $F_{k,\varepsilon}$, and the associated FY-loss. However, in most cases, clustering data can reasonably only be expected to be present as connectivity matrix. It is therefore necessary to alter the Fenchel-Young loss, and we introduce the following loss, which allows to work with partial information $p \in \mathcal{P}$ representing partial information about the unknown ground-truth $y$. Using this kind of "inf-loss" is common when dealing with partial label information (see, e.g., Cabannes et al., 2020).

**Definition 8** (Partial Fenchel-Young loss). *Let $F$ be a convex function, $L_{\mathsf{FY}}$ the associated Fenchel-Young loss, and for every $p \in \mathcal{P}$ a convex constraint subset $\mathcal{C}(p) \subseteq \mathcal{C}$.*

$$\bar{L}_{\mathsf{FY}}(\theta;p) = \min_{y \in \mathcal{C}(p)} L_{\mathsf{FY}}(\theta;y)\,.$$

This allows to learn from incomplete information about $y$. When we do not know its value, but only a subset of $\mathcal{C}(p) \subseteq \mathcal{C}$ to which it might belong, we can minimize the infimum of the FY losses that are consistent with the partial label information $y \in \mathcal{Y}(p)$.

**Proposition 1.** *When $F$ is the support function of a compact convex set $\mathcal{C}$, the partial Fenchel-Young loss (see Definition 8) satisfies*

1. *The loss is a difference of convex functions in $\theta$ given explicitly by*

$$\bar{L}_{\mathsf{FY}}(\theta;p) = F(\theta) - F(\theta;p)\,, \quad \text{where} \quad F(\theta;p) = \max_{y \in \mathcal{C}(p)} \langle y, \theta\rangle,$$

2. *The gradient with respect to $\theta$ is given by*

$$\nabla_\theta \bar{L}_{\mathsf{FY}}(\theta;p) = y^*(\theta) - y^*(\theta;p)\,, \quad \text{where} \quad y^*(\theta;p) = \operatorname*{argmax}_{y \in \mathcal{C}(p)} \langle y, \theta\rangle.$$

3. *The perturbed partial Fenchel-Young loss given by $\bar{L}_{\mathsf{FY},\varepsilon}(\theta;p) = \mathbf{E}[\bar{L}_{\mathsf{FY}}(\theta + \varepsilon Z;p)]$ satisfies*

$$\nabla_\theta \bar{L}_{\mathsf{FY},\varepsilon}(\theta;p) = y_\varepsilon^*(\theta) - y_\varepsilon^*(\theta;p)\,, \quad \text{where} \quad y_\varepsilon^*(\theta;p) = \mathbf{E}[\operatorname*{argmax}_{y \in \mathcal{C}(p)} \langle y, \theta + \varepsilon Z\rangle]\,.$$

Another possibility would be to define the partial loss as $\min_{y \in \mathcal{C}(p)} L_{\mathsf{FY},\varepsilon}(\theta;y)$, that is, the infimum of smoothed losses instead of the smoothed infimum loss $L_{\mathsf{FY},\varepsilon}(\theta;p)$ defined above. However, there is no direct method to minimizing the smoothed loss with respect to $y \in \mathcal{C}(p)$. Note that we have a relationship between the two losses:

**Proposition 2.** *Letting $L_{\mathsf{FY},\varepsilon}(\theta;y) = \mathbf{E}[L_{\mathsf{FY},\varepsilon}(\theta + \varepsilon Z;y)]$ and $\bar{L}_{\mathsf{FY},\varepsilon}$ as in Definiton 8, we have*

$$\bar{L}_{\mathsf{FY},\varepsilon}(\theta;p) \leq \min_{y \in \mathcal{C}(p)} L_{\mathsf{FY},\varepsilon}(\theta;y)\,.$$

The proofs of the above two propositions are detailed in the Appendix.

## 3.3 Applications to differentiable clustering

We apply this framework, as detailed in the following section and in Section 4, to clustering. This is done naturally by transposing notations and taking $\mathcal{C} = \mathcal{C}_k$, $\theta = \Sigma$, $y^* = A_k^*$, $p = M_\Omega$, $\mathcal{C}(p) = \mathcal{C}_k(M_\Omega)$, and $y^*(\theta;p) = A_k^*(\Sigma;M_\Omega)$. In this setting the perturbed partial Fenchel-Young loss satisfies

$$\nabla_\Sigma \bar{L}_{\mathsf{FY},\varepsilon}(\Sigma;M_\Omega) = A_{k,\varepsilon}^*(\Sigma) - A_{k,\varepsilon}^*(\Sigma;M_\Omega)\,.$$

We learn representations of a data that fit with clustering information (either complete or partial). As described above, we consider settings with $n$ elements described by their features $\mathbf{X} = X_1, \ldots, X_n$ in some feature space $\mathcal{X}$, and $M_\Omega$ some clustering information. Our pipeline to learn representations includes the following steps (see Figure 1)

*i)* Embed each $X_i$ in $\mathbb{R}^d$ with a parameterized model $v_i = \Phi_w(X_i) \in \mathbb{R}^d$, with weights $w \in \mathbb{R}^p$.

*ii)* Construct a similarity matrix from these embeddings, e.g. $\Sigma_{w,ij} = -\|\Phi_w(X_i) - \Phi_w(X_j)\|_2^2$.

*iii)* Stochastic optimization of the expected loss of $\bar{L}_{\mathsf{FY},\varepsilon}(\Sigma_{w,b}, M_{\Omega,b})$, using mini-batches of $\mathbf{X}$.

**Details.** To be more specific on each of these phases: *i)* We embed each $X_i$ individually with a model $\Phi_w$, using in our application neural networks and a linear model. This allows us to use learning through clustering as a way to learn representations, and to apply this model to other elements, for which we have no clustering information, or for use in other downstream tasks. *ii)* We focus on cases where the similarity matrix is the negative squared distances between those embeddings. This creates a connection between a model acting individually on each element, and a pairwise similarity matrix that can be used as an input for our differentiable clustering operator. This mapping, from $w$ to $\Sigma$, has derivatives that can therefore be automatically computed by backpropagation, as it contains only conventional opereations (at least when $\Phi_w$ is itself a conventional model, such as commonly used neural networks). *iii)* We use our proposed smoothed partial Fenchel-Young (Section 3.2) as the objective to minimize between the partial information $M_\Omega$ and $\Sigma$. The full-batch version would be to minimize $L_{\mathsf{FY},\varepsilon}(\Sigma_w, M_\Omega)$ as a function of the parameters $w \in \mathbb{R}^p$ of the model. We focus instead on a mini-batch formulation for two reasons: first, stochastic optimization with mini-batches is a commonly used and efficient method for generalization in machine learning; second, it allows to handle larger-scale datasets. As a consequence, the stochastic gradients of the loss are given, for a mini-batch $b$, by

$$\nabla_w \bar{L}_{\mathsf{FY},\varepsilon}(\Sigma_{w,b}; M_{\Omega,b}) = \partial_w \Sigma_w \cdot \nabla_\Sigma \bar{L}_{\mathsf{FY},\varepsilon}(\Sigma_{w,b}; M_{\Omega,b}) = \partial_w \Sigma_w \cdot \left( A^*_{k,\varepsilon}(\Sigma_w) - A^*_{k,\varepsilon}(\Sigma_w; M_{\Omega,b}) \right).$$

The simplicity of these gradients is due to our particular choice of smoothed partial Fenchel-Young loss. They can be efficiently estimated automatically, as described in Section 2.2, which results in a doubly stochastic scheme for the loss optimization.

## 4 Experiments

We apply our framework for learning through clustering in both a supervised and a semi-supervised setting, as illustrated in Figure 1. Formally, for large training datasets of size $n$, we either have access to a full cluster connectivity matrix $M_\Omega$ or a partial one (typically built by using partial label information, see below). We use this clustering information $M_\Omega$, from which mini-batches can be

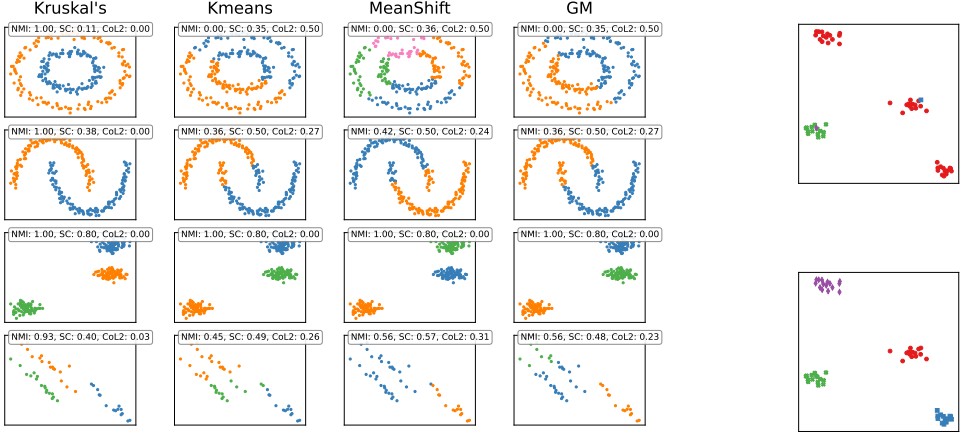

Figure 3: **(left)** Illustration of clustering methods on toy-data sets. **(right)** Using cluster information to learn a linear de-noising **(bottom)** of a noised signal **(top)**.

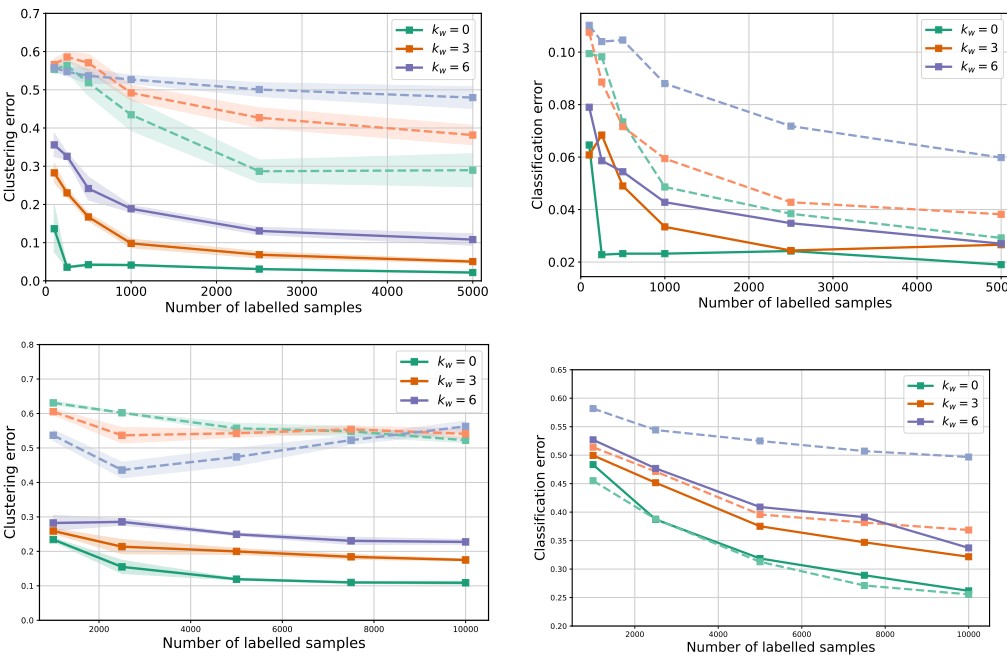

Figure 4: **Semi-supervised learning**: Performance of a CNN trained on partially labeled MNIST data **(top row)**, and a ResNet trained on partially labeled Cifar-10 data **(bottom row)**. Our trained model (full line) is evaluated on clustering **(left column)** and compared to a model entirely trained on the classification task (dashed line). Both models are evaluated on a down-stream classification problem i.e. transfer learning, via a linear probe **(right column)**.

extracted, as supervision. We minimize our partial Fenchel-Young loss with respect to the weights of an embedding model, and evaluate these embeddings in two main manners on a test dataset: first, by evaluating the clustering accuracy (i.e. proportion of correct coefficients in the predicted cluster connectivity matrix), and second by training a shallow model on a classification task (using clusters as classes) on a holdout set, evaluating it on a test set.

## 4.1 Supervised learning

We apply first our method to synthetic datasets - purely to provide an illustration of both our internal clustering algorithm, and of our learning procedure. In Figure 2, we show how the clustering operator that we use, based on spanning forests (i.e. single linkage), with Kruskal's algorithm is efficient on some standard synthetic examples, even when they are not linearly separable (compared to $k$-Means, mean-shift, Gaussian mixture-model). We also show that our method allows us to perform supervised learning based on cluster information, in a linear setting. For $X_{\text{signal}}$ consisting of $n = 60$ points in two dimensions consisting of data in four well-separated clusters (see Figure 2), we construct $X$ by appending two noise dimensions, such that clustering based on pairwise square distances between $X_i$ mixes the original clusters. We learn a linear de-noising transformation $X\theta$, with $\theta \in \mathbb{R}^{4 \times 2}$ through clustering, by minimizing our perturbed partial FY loss with SGD, using the known labels as supervision.

We also show that our method is able to cluster virtually all of some classical datasets. We train a CNN (LeNet-5 LeCun et al. (1998)) on mini-batches of size 64 using the partial Fenchel-Young loss to learn a clustering, with a batch-wise clustering precision of 0.99 for MNIST and 0.96 on Fashion MNIST. Using the same approach, we trained a ResNet (He et al., 2016) on CIFAR-10 (with some minor modifications to kernel size and stride), achieving a batch-wise clustering test precision of 0.93. The t-SNE visualization of the embeddings for the validation set are displayed in Figure 5. The experimental setups (as well as visualization of learnt clusters for MNIST), are detailed in the Appendix.

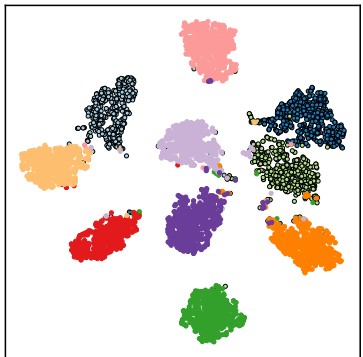 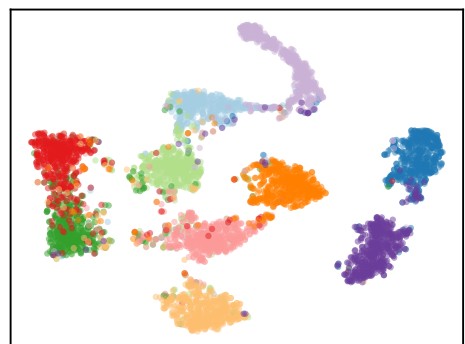

Figure 5: **(left)** t-SNE of learnt embeddings for the CNN trained on MNIST with $k_w = 3$ withheld classes (highlighted). **(right)** t-SNE of learnt embeddings for the ResNet trained on CIFAR-10.

## 4.2 Semi-supervised learning

We show that our method is particularly useful in settings where labelled examples are scarce, even in the particularly challenging case of having no labelled examples for some classes. To this end, we conduct a series of experiments on the MNIST (LeCun et al., 2010) and Cifar-10 (Krizhevsky et al., 2009) datasets, where we independently vary both the total number of labelled examples $n_\ell$, as well as the number of withheld classes for which no labelled examples are present in the training set $k_w \in \{0, 3, 6\}$. For MNIST we consider data sets with $n_\ell \in \{100, 250, 500, 1000, 2000, 5000\}$ labelled examples, and for Cifar-10 we consider $n_\ell \in \{1000, 2500, 5000, 7500, 10,000\}$.

We train the same embedding models using our partial Fenchel-Young loss with batches of size 64. We use $\varepsilon = 0.1$ and $B = 100$ for the estimated loss gradients, and optimize weights using Adam (Kingma and Ba, 2015).

In left column of Figure 4, we report the test clustering error (evaluated on mini-batches of the same size), for each of the models, and data sets, corresponding to the choice of $n_\ell$ and $k_w$. We compare the performance of each model to a baseline, consisting of the exact same architecture (plus a linear head mapping the embedding to logits), trained to minimize the cross entropy loss.

In addition, we evaluate all models on a down-stream (transfer-learning) classification task, by learning a linear probe on top of the frozen model, trained on hold-out data with all classes present. The results are depicted in the right hand column of Figure 4. Further information regarding the experimental details can be found in the Appendix.

We observe that learning through clustering allows to find a representation where class semantics are easily recoverable from the local topology. On Cifar-10, our proposed approach strikingly achieves a lower clustering error in the most challenging setting ($n_\ell = 1000$ labelled examples and $k_w = 6$ withheld classes) than the classification-based baseline in the most lenient setting ($n_\ell = 10,000$ labelled examples all classes observed). Importantly, this advantage is not limited to clustering metrics: learning through clustering can also lead to lower down-stream classification error, with the gap being most pronounced when few labelled examples are available.

Moreover, besides this pronounced improvement in data efficiency, we found that our method is capable of clustering classes for which no labelled examples are seen during training (see Figure 5, left). Therefore, investigating potential applications of learning through clustering to zero-shot and self-supervised learning are promising avenues for future work.

## Acknowledgements

We thank Jean-Philippe Vert for discussions relating to this work, Simon Legrand for conversations relating to the CLEPS computing cluster (Cluster pour l'Expérimentation et le Prototypage Scientifique), and Matthieu Blondel for providing references regarding learning via Kirchhoff's theorem. We also thank the reviewers and Kyle Heuton, for providing helpful corrections for the camera-ready paper. We also acknowledge support from the French government under the management of the Agence Nationale de la Recherche as part of the "Investissements d'avenir" program, reference ANR19-P3IA-0001 (PRAIRIE 3IA Institute), as well as from the European Research Council (grant SEQUOIA 724063), and a monetary *don* from Google.

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

## 5 Broader impact

This submission focuses on foundational and exploratory work, with application to general machine learning techniques. We propose methods to extend the range of operations that can be used in end-to-end differentiable systems, by allowing to include clustering as a differentiable operation in machine learning models. We do not foresee societal consequences that are specifically related to these methods, beyond those that are associated with the field in general.

## 6 Reproducibility and Licensing

Our experiments use data sets that are already open-sourced and cited in the references. All the code written for this project and used to implement our experiments is available at:

https://github.com/LawrenceMMStewart/DiffClust_NeurIPS2023,

and distributed under a Apache 2.0. license.

## 7 Limitations

At present, our implementation of Kruskal's algorithm is incompatible with processing very large batch sizes at train time. However, as our method can use mini-batches, it is suitable for training deep learning models on large data sets. For example, we use a batch size of 64 (similar to that for standard classification models) for all experiments detailed in our submission. Other methods that focus on cluster assignment, rather than pairwise clustering, often have an $n \times k$ parametrization, whose efficiency will depend on the comparison between $n$ (batch size) and $k$ (number of clusters).

At inference time this is not the case, since gradients need not be back-propagated hence, any implementation of Kruskal's algorithm can be used such as the union-find implementation. Faster GPU compatible implementations of Kruskal's algorithm for training are certainly possible ; improvements could be made using a binary heap or a GPU parallelized merge sort. In many learning experiments the batch sizes are typically of values $\{32, 64, 128, 256, 512\}$, so our current implementation is sufficient, however, for some specific applications (such as bio-informatics experiments requiring tens of thousands of points in a batch), it would be necessary to improve upon the current implementation. This is mainly an engineering task relating to data-structures and GPU programming, and is beyond the scope of this paper.

Finally, our clustering approach solves a linear program whose argmax solution is a matroid problem with a greedy single-linkage criterion. Whilst we saw that this method is effective for differentiable clustering in both the supervised and semi-supervised setting, there may well be other linkage criteria which also take the form of LP's but whom are more robust to outliers and noise in the train dataset. This line of work is outside the scope of the paper, but is closely related and could help improve our differentiable spanning forests framework.

## 8 Proofs of Technical Results

*Proof of Proposition 1.* We show these properties successively

1) We have by Definition 8
$$\bar{L}_{\mathsf{FY}}(\theta; p) = \min_{y \in \mathcal{C}(p)} L_{\mathsf{FY}}(\theta; y) \,.$$

We expand this
$$\bar{L}_{\mathsf{FY}}(\theta; p) = \min_{y \in \mathcal{C}(p)} \left\{ F(\theta) - \langle y, \theta \rangle \right\} = F(\theta) - \max_{y \in \mathcal{C}(p)} \langle y, \theta \rangle \,.$$

As required, this implies, following the definitions given
$$\bar{L}_{\mathsf{FY}}(\theta; p) = F(\theta) - F(\theta; p) \,, \quad \text{where} \quad F(\theta; p) = \max_{y \in \mathcal{C}(p)} \langle y, \theta \rangle,$$

2) By linearity of derivatives and 1) above, we have
$$\nabla_\theta \bar{L}_{\mathsf{FY}}(\theta; p) = \nabla_\theta F(\theta) - \nabla_\theta F(\theta; p) = y^*(\theta) - y^*(\theta; p) \,, \quad \text{where} \quad y^*(\theta; p) = \operatorname*{argmax}_{y \in \mathcal{C}(p)} \langle y, \theta \rangle,$$

Since the argmax of a constrained linear optimization problem is the gradient of its value. We note that this property holds almost everywhere (when the argmax is unique), and almost surely for costs with positive, continuous density, which we always assume (e.g. see the following).

3) By linearity of expectation and 1) above, we have

$$\nabla_\theta \bar{L}_{\mathsf{FY},\varepsilon}(\theta;p) = \nabla_\theta \mathbf{E}[F(\theta+\varepsilon Z) - F(\theta+\varepsilon Z;p)] = \nabla_\theta F_\varepsilon(\theta) - \nabla_\theta F_\varepsilon(\theta;p) = y_\varepsilon^*(\theta) - y_\varepsilon^*(\theta;p)\,,$$

using the definition

$$y_\varepsilon^*(\theta;p) = \mathbf{E}[\operatorname*{argmax}_{y \in \mathcal{C}(p)} \langle y, \theta + \varepsilon Z \rangle]\,.$$

$\square$

*Proof of Proposition 2.* By Jensen's inequality and the definition of the Fenchel-Young loss:

$$\begin{aligned}
\bar{L}_{\mathsf{FY},\varepsilon}(\theta;p) &= \mathbf{E}[\min_{y \in \mathcal{C}(p)} L_{\mathsf{FY}}(\theta+\varepsilon Z;y)] \\
&\leq \min_{y \in \mathcal{C}(p)} \mathbf{E}[L_{\mathsf{FY}}(\theta+\varepsilon Z;y)] \\
&= \min_{y \in \mathcal{C}(p)} F_\varepsilon(y) - \langle \theta, y \rangle \\
&\leq \min_{y \in \mathcal{C}(p)} L_{\mathsf{FY},\varepsilon}(\theta;y)\,.
\end{aligned}$$

$\square$

# 9 Algorithms for Spanning Forests

As mentioned in Section 2, both $A_k^*(\Sigma)$ and $M_k^*(\Sigma)$ are calculated using Kruskal's algorithm (Kruskal, 1956). Our implementation of Kruskal's is tailored to our use: we first initialize both $A_k^*(\Sigma)$ and $M_k^*(\Sigma)$ as the identity matrix, and then sort the upper triangular entries of $\Sigma$. We build the maximum-weight spanning forest in the usual greedy manner, using $A_k^*(\Sigma)$ to keep track of edges in the forest and $M_k^*(\Sigma)$ to check if an edge can be added without creating a cycle, updating both matrices at each step of the algorithm. Once the forest has $k$ connected components, the algorithm terminates. This is done by keeping track of the number of edges that have been added at any time.

We remark that our implementation takes the form as a single loop, with each step of the loop consisting only of matrix multiplications. For this reason it is fully compatible with auto-differentiation engines, such as JAX (Bradbury et al., 2018), Pytorch (Paszke et al., 2019) and TensorFlow (Abadi et al., 2016), and suitable for GPU/TPU acceleration. Therefore, our implementation differs from that of the standard Kruskal's algorithm, which used a disjoint union-find data structure (and hence is not compatible with auto-differentiation frameworks).

## 9.1 Constrained Spanning Forests

As an heuristic way to solve the constrained problem detailed in Section 5, we make the modifications below to our implementation of Kruskal's, under the assumption that $M_\Omega$ represents *valid* clustering information (i.e. with no contradiction):

1. **Regularization** (Optional) : It is possible to bias the optimization problem over spanning forests to encourage or discourage edges between some of the nodes, according to the clustering information. Before performing the sort on the upper-triangular of $\Sigma$, we add a large value to all entries of $\Sigma_{ij}$ where $(M_\Omega)_{ij} = 1$, and subtract this same value from all entries of $\Sigma_{ij}$ where $(M_\Omega)_{ij} = 1$. Entries $\Sigma_{ij}$ corresponding to where $(M_\Omega)_{ij} = \star$ are left unchanged. This biasing ensures that any edge between points that are constrained to be in the same cluster will always be processed before unconstrained edges. Similarly, any edge between points that are constrained to not be in the same cluster, will be processed after unconstrained edges. In most cases, i.e. when all clusters are represented in the partial information, such as when $\Omega = [n] \times [n]$ (full information), this is not required to solve the constrained linear program, but we have found that this regularization was helpful in practice.

2. **Constraint enforcement** : We ensure that adding an edge does not violate the constraint matrix. In other words, when considering adding the edge $(i, j)$ to the existing forest, we check that none of the points in the connected component of $i$ are forbidden from joining any of the points in the connected component of $j$. This is implemented using further matrix multiplications and done alongside the existing check for cycles. The exact implementation is detailed in our code base.

## 10 Existing Literature on Differentiable Clustering

As discussed in Section 1, there exists many approaches which use clustering during gradient based learning, but these approaches typically use clustering in an offline fashion in order to assign labels to points. The following methods aim to learn through a clustering step (i.e. gradients back-propagate through clustering):

Yang et al. (2017) use a bi-level optimization procedure (alternating between optimizing model weights and centroid clusters). They reported attaining 83% label-wise clustering accuracy on MNIST using a fully-connected deep network. Our method differs from this approach as it is allows for end-to-end online learning.

Genevay et al. (2019) cast k-means as an optimal transport problem, and uses entropic regularization for smoothing. Reported a 85% accuracy on MNIST and 25% accuracy on CIFAR-10 with a CNN.

## 11 Additional Experimental Information

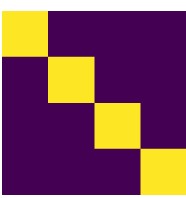 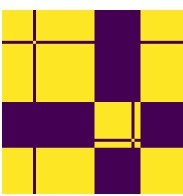 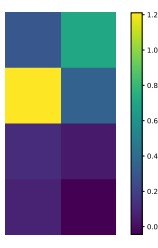

Figure 6: Leftmost figure: $M_{signal}$, Center Figure: $M_4^*(\Sigma)$, Rightmost Figure: $\theta$ after training.

We provide the details of the synthetic denoising experiment depicted in Figure 2 and described in Section 4.1. We create the signal data $X_{\text{signal}} \in \mathbb{R}^{60 \times 2}$ by sampling iid. from four isotropic Gaussians (15 points coming from each of the Gaussians) each having a standard deviation of 0.2. We randomly sample the means of the four Gaussians; for the example seen in Section 4.1 the sampled means were:

$$\begin{pmatrix} 0.97627008 & 4.30378733 \\ 2.05526752 & 0.89766366 \\ -1.52690401 & 2.91788226 \\ -1.24825577 & 7.83546002 \end{pmatrix}.$$

Let $\Sigma_{\text{signal}}$ be the pairwise euclidean similarity matrix corresponding to $X_{\text{signal}}$, and furthermore let $M_{\text{signal}} := M_4^*(\Sigma_{\text{signal}})$ be the equivalence matrix corresponding to the signal ($M_{\text{signal}}$ will be the target equivalence matrix to learn).

We append an additional two 'noise dimensions' to $X_{\text{signal}}$ in order to form the train data $X \in \mathbb{R}^{60 \times 4}$, where the noise entries were sampled iid from a continuous unit uniform distribution. Similarly letting $\Sigma$ be the pairwise euclidean similarity matrix corresponding to $X$, we calculate $M_4^*(\Sigma) \neq M_{\text{signal}}$. Both the matrices $M_{\text{signal}}$ and $M_4^*(\Sigma)$ are depicted in Figure 6 ; we remark that adding the noise dimensions leads to most points being assigned one of two clusters, and two points being isolated alone in their own clusters. We also create a validation set of equal size (in exactly the same manner as the train set), to ensure the model has not over-fitted to the train set.

The goal of the experiment is to learn a linear transformation of the data that recovers $M_{\text{signal}}$ i.e. a denoising, by minimizing the partial loss. There are multiple solutions to this problem, the most obvious being a transformation that removes the last two noise columns from $X$:

$$\theta^* = \begin{pmatrix} 1 & 0 \\ 0 & 1 \\ 0 & 0 \\ 0 & 0 \end{pmatrix}, \quad \text{for which} \quad X\theta^* = X_{\text{signal}}$$

For any $\theta \in \mathbb{R}^{4 \times 2}$, we define $\Sigma_\theta$ to be the pairwise similarity matrix corresponding to $X\theta$, and $M_4^*(\Sigma_\theta)$ to its corresponding equivalence matrix. Then the problem can be summarized as:

$$\min_{\theta \in \mathbb{R}^{4 \times 2}} \bar{L}_{\text{FY},\varepsilon}(\Sigma_\theta, M_{\text{signal}}). \tag{1}$$

We initialized $\theta$ from a standard Normal distribution, and minimized the partial loss via stochastic gradient descent, with a learning rate of $0.01$ and batch size $32$.

For perturbations, we took $\varepsilon = 0.1$ and $B = 1000$, where $\varepsilon$ denotes the noise amplitude in randomized smoothing and $B$ denotes the number of samples in the Monte-Carlo estimate. The validation clustering error converged to zero after $25$ gradient batches. We verify that the $\theta$ attained from training is indeed learning to remove the noise dimensions (see Figure 6).

## 11.1 Supervised Differentiable Clustering

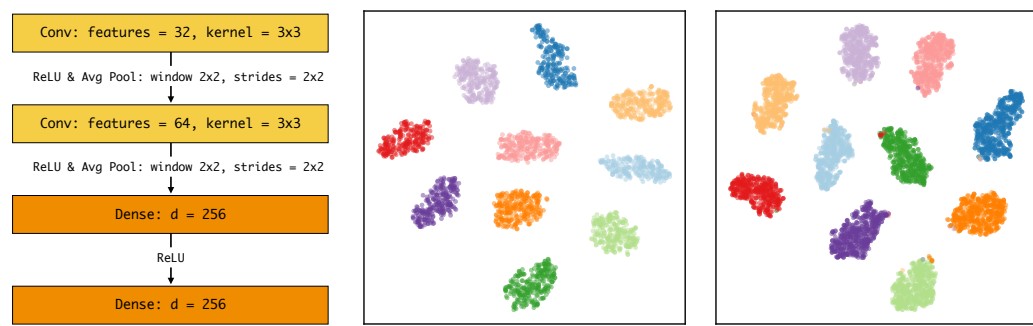

Figure 7: **(left)** Architecture of the CNN, **(middle)** t-SNE visualization of train data embeddings, **(right)** tSNE visualization of validation data embeddings.

As mentioned in Section 4.1, our method is able to cluster classical data sets such as MNIST and Fashion MNIST. We trained a CNN with the LeNet-5 architecture LeCun et al. (1998) using our proposed partial loss as the objective function. The exact details of the CNN architecture are depicted in Figure 7. For this experiment and all experiments described below, we trained on a single Nvidia V100 GPU ; training the CNN with our proposed pipeline took $< 15$ minutes.

The model was trained for 30k gradient steps on mini-batches of size $64$. We used the Adam optimizer (Kingma and Ba, 2015) with learning rate $\eta = 3 \times 10^{-4}$, momentum parameters $(\beta_1, \beta_2) = (0.9, 0.999)$, and an $\ell_2$ weight decay of $10^{-4}$. We validated / tested the model using the zero-one error between the true equivalence matrix and the equivalence matrix corresponding to the output of the model. We used an early stopping of 10k steps (i.e. training was stopped if the validation clustering error did not improve over 10k steps). For efficiency (and parallelization), we also computed this clustering error batch-wise with batch-size $64$. As stated in Section 4.1, we attained a batch-wise clustering precision of $0.99$ for MNIST and $0.96$ on Fashion MNIST.

The t-SNE visualizations of the embedding space of the model trained on MNIST for a collection of train and validation data points are depicted in Figure 7. It can be seen that the model has learnt a distinct cluster for each of the ten classes.

In similar fashion, we trained a ResNet (He et al., 2016) on the Cifar-10 data set. The exact model architecture is similar to that of ResNet-50, but with minor modifications to the input convolutions for compatibility with the dimensions of Cifar images, and is detailed in the code base.

The training procedure was identical to that of the CNN, except the model was trained for 75k steps (with early stopping), and used the standard data augmentation methods for Cifar-10, namely: a

combination of four-pixel padding, random flip followed by a random crop. As mentioned in Section 4.1, the model achieved a batch-wise clustering test precision of $0.933$.

## 11.2  Semi-Supervised Differentiable Clustering

As mentioned in Section 4.2, we show that our method is particularly useful in settings where labelled examples are scarce, even in the particularly challenging case of having no labelled examples for some classes. Our approach allows a model to leverage the semantic information of unlabeled examples when trying to predict a target equivalence matrix $M_\Omega$ ; this is owing to the fact that the prediction of a class for a single point depends on the values of all other points in the batch, which is in general not the case for more common problems such as classification and regression.

To demonstrate the performance of our approach, we assess our method on two tasks:

1. **Semi-Supervised Clustering:** Train a model to learn an embedding of the data which leads to a good clustering error. We can compare our methodology to that of a baseline model trained using the cross-entropy loss. This is to check that our model has leveraged information from the unlabeled data and that our partial loss is indeed leading to good clustering performance.

2. **Downstream Classification:** Assess the trained model's capacity to serve as a backbone in a downstream classification task (transfer learning), where its weights are frozen and a linear layer is trained on top of the backbone.

We describe our data processing for both of these tasks below.

### 11.2.1  Data Sets

In our semi-supervised learning experiments, we divided the standard MNIST and Cifar-10 train splits in the following manner:

- We create a balanced hold-out data set consisting of 1k images (100 images from each of the 10 classes). This hold-out data set will be used to assess the utility of the frozen clustering model on a downstream classification problem.

- From the remaining 59k images, we select a labeled train set of $n_\ell$ points (detailed in Section 4.2). Our experiments also vary $k_w \in \{0, 3, 6\}$, the number of digits to withhold all labels from. For example, if $k_w = 3$, then the labels for the images corresponding to digits $\{0, 1, 2\}$ will never appear in the labeled train data.

### 11.2.2  Semi-Supervised Clustering Task

For each of the choices of $n_\ell$ and $k_w$, we train the architectures described in Section 11.1 using the following two approaches:

1. **Ours:** The model is trained on mini-batches, where half the batch is labeled data and half the batch is unlabeled data (i.e. a semi-supervised learning regime), to minimize the partial loss.

2. **Baseline:** The baseline model shares the same architecture as that described in Section 11.1, but with an additional dense layer with output dimension 10 (the number of classes). We train the model using mini-batches consisting of labeled points, minimizing the cross-entropy loss. The training regime is fully-supervised learning (classification). The baseline backbone refers to all of the model, minus the dense output layer.

Both models were trained with mini-batches of size 64, with points sampled uniformly without replacement. All hyper-parameters and optimization metrics were identical to those detailed in Section 4.1. For MNIST, we repeated training for each model with five different random seeds $s \in [5]$ (and with three random seeds $s \in [3]$ for Cifar), in order to report population statistics on the clustering error.

### 11.2.3 Transfer-Learning: Downstream Classification

In this task both models are frozen, and their utility as a foundational backbone is assessed on a downstream classification task using the hold-out data set. We train a linear (a.k.a dense) layer on top of both models using the cross-entropy loss. We refer to this linear layer as the downstream head. Training this linear head is equivalent to performing multinomial logistic regression on the features of the model.

To optimize the weights of the linear head we used the SAGA optimizer (Defazio et al., 2014). The results are depicted in Figure 4. It can be seen that training a CNN backbone using our approach with just 250 labels leads to better downstream classification performance than the baseline trained with 5000 labels. It is worth remarking that the baseline backbone was trained on the same objective function (cross-entropy) and task (classification) as the downstream problem, which is not the case for the backbone corresponding to our approach. This highlights how learning 'cluster-able embeddings' and leveraging unlabeled data can be desirable for transfer learning.

