# OpenReview forum: "Differentiable Clustering with Perturbed Spanning Forests"
_NeurIPS.cc/2023/Conference — NeurIPS 2023 poster_

### Official Review · Reviewer_sM5n · 2023-07-02

**Soundness:** 3 good
**Presentation:** 3 good
**Contribution:** 3 good
**Rating:** 6
**Confidence:** 4

**Summary:**

This paper proposes a differentiable clustering method based on a stochastic perturbation of mnimum-weight spanning forests. They demonstrate its performance on several datasets in supervised and semi-supervised settings.

**Strengths:**

This paper introduces a differentiable clustering method based on linear programs for minimum-weight spanning forests.
Their method starts with formulating the clustering method as spanning forest problem, which admits an LP (linear programming) formulation. Then, they make the parametrization differentiable using stochastic perturbation. They also propose a novel partial Fenchel-Young loss, which allows for learning through clustering in weakly supervised problem.

The technical part of this paper is strong. The idea of partial Fenchel-Young loss is novel. The theoretical part is well written and not hard to follow.

**Weaknesses:**

For experiment part,

- it might be better to elaborate more on the settings and what and how $M_\Omega$ is given in these settings so readers can better understand the tasks.

- Comparison with other methods is lacking: maybe some SOTA nondifferentiable clustering methods

- Since the goal is to learn $M$, an $n\times n$ matrix, would it be some computational efficiency/scalability issue of the proposed method when $n$ is large? Other methods might have more condensed parametrization than the proposed framework?

**Questions:**

See Weaknesses

**Limitations:**

See Weaknesses

---

> ### Author Rebuttal · Authors · 2023-08-09
>
> Thank you very much for your positive evaluation of our work, as well as the interesting points that you raise. We have taken this opportunity to address them in the following manner.
>
> ### *Elaborating on the setting*
>
> We have added a remark clarifying how $M_\Omega$ is given, from label information. To summarize, for all pairs of elements $(i, j)$ who both have label information, $M_{\Omega, ij}$ is equal to $1$ if they have the same label (must-link) and $0$ if they have a different one (must-not-link). The coefficient $M_{\Omega, ij}$ is ‘$*$’ (no information) if one of the two labels is not known.
>
> ### *Comparison with nondifferentiable clustering methods*
>
> We have added a quantitative comparison for different clustering methods (k-means, Gaussian Mixture Model), both on the standard synthetic data showcased in Figure 3 of the submission, as well as on the CIFAR-10 dataset, using pre-trained features.
>
> The results on the standard synthetic data showcased in Figure 3 have been added to our manuscript - see Figure 1.(a) in the attached document.
>
> The results for CIFAR-10 are as follows, they have been added to our manuscript
>
> +++++
>
> CIFAR-10 (ResNet50 for embeddings) [metric: L2 distance in coincidence matrix]:
>
> Differentiable clustering: 0.097 (ours)
>
> Classification embeddings + k-means: 0.104 (non-differentiable)
>
> Classification embeddings + GMM: 0.107 (non-differentiable)
>
> +++++
>
> This allows us to highlight that our method not only allows us to perform differentiable clustering, it allows us to learn good representations of the data.
>
> ### *Parametrization of our method*
>
> Our method indeed scales quadratically in $n$, similarly to attention mechanisms. However, as our method can use mini-batches, it is suitable for training deep learning models on large data sets. For example, we use a batch size of 64 (similar to that for standard classification models) for all experiments detailed in our submission.
>
> Other methods that focus on cluster assignment, rather than pairwise clustering, often have an $n \times k$ parametrization, which can be smaller or larger, depending on the comparison between $n$ (batch size) and $k$ (number of clusters). We have added a remark to clarify both of these points.

---

> > ### Comment · Reviewer_sM5n · 2023-08-19
> > **Thanks for your response**
> >
> > Dear Authors,
> >
> > Thank you for taking time to address my concerns, and sorry for the late reply. As for $M_{\Omega}$, I understood the setting of must-link and must-not-link labels correspond to 1 and 0 entries in $M_\Omega$. I guess my question is more like how this info is given from datasets --- is this provided by the datasets or is it manually determined or is it randomly generated? For example, for the datasets used in the experiments, like standard synthetic data,  as well as the CIFAR or MNIST/Fashion-MNIST data.
> >
> > I am good with other points. Thanks!

---

> > > ### Author Response · Authors · 2023-08-20
> > >
> > > Thank you for your reply.
> > >
> > > Regarding your question about $M_\Omega$: In our experiments, this is done in two parts
> > >
> > > - First, artificially removing the label information for some of the instances. This is done randomly ahead of any training. The set of elements with no label information is fixed and stays the same. This is meant to simulate a dataset where some information is indeed missing. In practice, this is not necessary, but done in our experiments of the method to evaluate the impact of number of labeled elements/classes.
> > >
> > > - Then, at every batch, this information is used to create $M_\Omega$, going from same class / different class information to must-link / must-not-link / no information, as described in our reply above.
> > >
> > > We will add a remark to clarify this in our manuscript, thank you for the opportunity to do so.
> > >
> > > Other kind of partial clustering information can of course be used if it is available: our method can tackle any $M_\Omega$ that is consistent with some true cluster information.

---

> > > > ### Comment · Reviewer_sM5n · 2023-08-20
> > > >
> > > > Thanks for your explanation. Now it is more clear, and I will raise my score. Thanks again for your efforts and the good work.

---

### Official Review · Reviewer_ukVi · 2023-07-06

**Soundness:** 3 good
**Presentation:** 3 good
**Contribution:** 3 good
**Rating:** 8
**Confidence:** 3

**Summary:**

The paper presents a novel method for differentiable clustering in combination with a novel loss derived from Fenchel-Young losses.

**Strengths:**

The paper proposes a new concept for differentiable clustering.
Differentiable clustering has been a topic with only few applicable methods so far, so it is great to see a new and at the same time strong method being proposed in this space.

The method is sound and the utility of the method is experimentally demonstrated.

**Weaknesses:**

It would have been great to see a stronger experiment. Maybe CIFAR-10 / -100? Or for self-supervised learning on STL or ImageNet?
I would increase my score for a stronger experiment.


The main aspect of being able to transmit gradients backward through a clustering operation, a good representation of data points, as computed e.g. by some neural network, can be learned. That is, the original data, which might be difficult to cluster, maybe even due to its dimensionality, is not clustered directly, but a learnable representation of it is clustered, and this representation is informed by the gradients transmitted backwards through the clustering algorithm. I believe this should be emphasized more in the introduction.

The clustering function \pi is usually described by a k x n matrix U that is called "partition matrix". The matrix that is called a "cluster membership matrix" here (not a good name, because it does not state which data points belong to which clusters -- that's the partition matrix) is usually known as "coincidence matrix" or "cluster connectivity matrix" (e.g. in the context of relative cluster evaluation measures -- it states for each set of data points whether they are assigned to the same cluster or not). I would recommend using this standard terminology. The entries of a coincidence matrix M can be computed from the entries of a partition matrix U as M_jl = \sum_{i=1}^k U_ij U_il. Note that this works also in the fuzzy or membership degree case (i.e. U_ij \in [0,1]). In this sense, Def. 4 does not specify membership constraints, but rather coincidence constraints or connectivity constraints.

The terms "must link" and "must not link" that occur in the caption of Figure 1 could have been introduced after Def. 4, as "must link": M_\Omega,ij = 1, "must not link": M_\Omega,i,j = 0. In this sense M is a "cluster connectivity matrix" (see above).

The connection to single linkage hierarchical agglomerative clustering with cluster merging carried out until k clusters remain could have been pointed out.

126: a n x n -> an n x n


[A] seems to be a relevant but missing reference.

[A] Struminsky, K., Gadetsky, A., Rakitin, D., Karpushkin, D., and Vetrov, D. P. Leveraging recursive gumbel-max trick for approximate inference in combinatorial spaces. Advances in Neural Information Processing Systems, 2021.

**Questions:**

see above.

**Limitations:**

The experimental evaluation is limited.

---

> ### Author Rebuttal · Authors · 2023-08-09
>
> Thank you very much for your positive evaluation of our work, as well as the interesting points that you raise. We have taken this opportunity to address them in the following manner.
>
>  ### *Scaling to more complicated datasets*
>
> Thank you very much for your suggestion, and the indication that you would raise your score. As suggested, we have added experiments on the CIFAR-10 dataset for both the fully supervised and semi-supervised settings (as well as a downstream classification transfer learning task, as in Figure 4 of the submission).
>
> For supervised clustering, we obtain a clustering error of 6.7% on the test set, in line with our results on MNIST and Fashion-MNIST, we also provide in Figure 1.(b) the t-SNE of embeddings obtained in this manner in the feature space.
>
> For semi-supervised clustering, the results are in Figure 2 in the attached document and have been included in the manuscript (analogous to Figure 4 on MNIST in our original submission). We observe analogous trends for clustering error to those detailed in Sections 4.2 (& 10.2) of the submission, with our model outperforming the baseline. The model attained a comparable classification error via a linear head (downstream transfer learning), to that of a baseline classification model for k=0, and for k=3, 6 outperformed the baseline.
>
> ### *On learning representations as the main aspect of being able to transmit gradients backward through a clustering operation*
>
> Thank you very much for bringing this up. Indeed, this is a very interesting way to present our work, and an important point, that we had not clearly expressed. We have added a remark highlighting this point in the introduction.
>
> ### *Notation and presentation*
>
> We have made all of the suggested terminology and presentation changes.
>
> ### *Suggested reference*
>
> Thank you for this very interesting reference, we have added a citation to it, as well as discussion of this work.

---

> > ### Comment · Reviewer_ukVi · 2023-08-15
> >
> > Thank you for your response and for addressing all of my concerns. Accordingly, I raise my score.
> >
> > In the vein of learning representations with differentiable grouping, [B] could be a relevant recent reference.
> >
> > [B] Nina Shvetsova et al. "Learning by Sorting: Self-supervised Learning with Group Ordering Constraints", ICCV 2023

---

> > > ### Author Response · Authors · 2023-08-15
> > >
> > > Thank you for your positive response to our rebuttal, and for the relevant reference on representation learning via grouping (ICCV 2023). We have added a citation to it, as well as a discussion of the work.

---

### Official Review · Reviewer_mX78 · 2023-07-10

**Soundness:** 3 good
**Presentation:** 3 good
**Contribution:** 3 good
**Rating:** 6
**Confidence:** 2

**Summary:**

This paper introduces a differentiable clustering method based on minimum-weight spanning forests.
This method can be integrated into end-to-end trainable pipelines and handle datasets with high noise and challenging geometries.
The key idea is to smooth the combinatorial clustering operation by taking expectations over stochastic perturbations of the input similarity matrix.
By taking expectations over these perturbations, gradients can be computed and backpropagated through the clustering process, facilitating end-to-end training.
Experiments on both supervised and semi-supervised tasks demonstrate the effectiveness of the proposed method.

**Strengths:**

1. The authors propose an effective method to address a challenging problem: learning through clustering.
The proposed method is well-motivated and reasonable, and it has the potential to be applied in a wide range of applications.
This paper addresses the key challenges of incorporating combinatorial operations like clustering into neural network pipelines that can be optimized through gradient descent. The perturbed proxy for discrete operators can address the major issue of piece-wise constant by attaining useful properties like smoothness and computational traceability.

2. The paper is well-written and easy to follow. Detailed remarks are provided for readers to better understand the definitions.

**Weaknesses:**

My major concern is about the evaluation.

1. The proposed method is only evaluated on small-scale datasets such as MNIST. It remains unclear whether the proposed method can be scaled to large datasets.

2. The authors did not compare the proposed method with other differentiable clustering methods. Therefore, it remains unclear whether the proposed method is more efficient than other baselines.

**Questions:**

Please check the weakness.

**Limitations:**

The authors claim that the proposed method can be used to handle datasets with high noise, but this claim is not verified in the experiments.

---

> ### Author Rebuttal · Authors · 2023-08-09
>
> Thank you very much for your positive evaluation of our work, as well as the interesting points that you raise. We have taken this opportunity to address them in the following manner.
>
> ### *Scaling to more complicated datasets*
>
> As our method can use mini-batches, it is suitable for training deep learning models on large datasets (we use a batch size of 64 for all experiments detailed in our submission).
>
> We have added experiments on the CIFAR-10 dataset for both the fully supervised and semi-supervised settings (as well as a downstream classification transfer learning task, as in Figure 4 of the submission).
>
> For supervised clustering, we obtain a clustering error of 6.7% on the test set, in line with our results on MNIST and Fashion-MNIST, we also provide in Figure 1.(b) the t-SNE of embeddings obtained in this manner in the feature space.
>
> For semi-supervised clustering, the results are in Figure 2 in the attached document and have been included in the manuscript (analogous to Figure 4 on MNIST in our original submission). We observe analogous trends for clustering error to those detailed in Sections 4.2 (& 10.2) of the submission, with our model outperforming the baseline. The model attained a comparable classification error via a linear head (downstream transfer learning), to that of a baseline classification model for k=0, and for k=3, 6 outperformed the baseline.
>
>
> ### *Comparison with other differentiable clustering methods*
>
> We have added a comparison between our method’s results on MNIST and CIFAR-10 and those reported by [Yang et al. 2017] and [Genevay et al. 2019], because they were the most comparable to our own experimental evaluations. As a reminder, using our methodology, we obtain a clustering error of 1.0% on MNIST and 6.7% on CIFAR-10 (and 4.0% on Fashion-MNIST).
> A remark describing this comparison has been included in the manuscript.
>
> + [Yang et al. 20177] - Towards K-means-friendly Spaces: Simultaneous Deep Learning and Clustering, ICML 2017
>
> Uses an bi-level optimization procedure (alternating between optimizing model weights and centroid clusters). They reported attaining 83% label-wise clustering accuracy on MNIST using a fully-connected deep network.
>
> + [Genevay et al. 2019] -  ​​Differentiable Deep Clustering with Cluster Size Constraints, 2019
>
> Casts k-means as an optimal transport problem, and uses entropic regularization for smoothing. Reported a 85% accuracy on MNIST and 25% accuracy on CIFAR-10 with a CNN.

---

> > ### Comment · Reviewer_mX78 · 2023-08-18
> >
> > Thank the authors for addressing my concern by including experiments on the CIFAR-10 dataset. Although CIFAR-10 is still a small dataset, it is more complicated than MNIST and Fashion-MNIST. I prefer to increase my score to 6.
> > It would be greatly appreciated if the authors could include the new results in the revised manuscript

---

> > > ### Author Response · Authors · 2023-08-20
> > >
> > > Thank you very much for your comment, and for increasing your score, we are glad that we were able to address your concerns. We have added these results to our work, and they will be in the revised manuscript.

---

### Official Review · Reviewer_cJCp · 2023-07-18

**Soundness:** 3 good
**Presentation:** 3 good
**Contribution:** 3 good
**Rating:** 5
**Confidence:** 2

**Summary:**

The paper presents a differentiable clustering approach based on k-spanning forests. It formulates the clustering assignment as an NxN membership matrix while introducing partial Fenchel-Young losses to optimize the between the cluster constraint and similarity/affinity matrix. The study demonstrates the applicability of the proposed approach to both supervised and semi-supervised learning for representation learning through clustering. The results indicate the following: (1) On several synthetic datasets, the proposed method generated qualitatively better clusters compared to baseline clustering methods including k-means, MeanShift, and Gaussian mixture model. (2) In supervised/semi-supervised representation learning experiments, the proposed method effectively learns a robust representation based on the labels (cluster constraint), outperforming the classification approach on the MNIST/Fashion-MNIST datasets.

**Strengths:**

- The proposed approached based on k-spanning tree and the partial Fenchel-Young losses for clustering is technical sound.
- The proposed approach enable end-to-end optimization for representation learning through clustering which is a novel and interesting direction. Experimental results on MNIST also shows that interestingly the method was able to outperform classification loss on semi-supervised tasks especially on low-data regime.

**Weaknesses:**

- Experimental results are only limited to toy datasets (synthesized data, MNIST), it would be interesting to see how the methods expands to real world data such as ImageNet.
- Comparison between traditional clustering methods are also limited to qualitative comparison, it would be good to have more quantitive  comparison between different clustering methods and demonstrate the benefits of the proposed clustering method.

**Questions:**

- How does the proposed clustering methods compared to traditional clustering algorithm (e.g. k-means) on real worlds data? The paper only present qualitative experiments on toy dataset, it would be more interesting to know how it compare quantitatively on real world dataset.
- How does the proposed method perform on more complicate dataset such as ImageNet compare to traditional classification approach? It would make the paper stronger if we could show that how the proposed method could be incorporate into ML pipeline to demonstrate benefits over more complicate dataset.

**Limitations:**

The authors have listed the possible broader impact and limitations (e.g. batch size and current implementation efficiency) in the supplemental materials.

---

> ### Author Rebuttal · Authors · 2023-08-09
>
> Thank you very much for your positive evaluation of our work, as well as the interesting points that you raise. We have taken this opportunity to address them in the following manner.
>
> ### *Comparison with other clustering algorithms*
>
> We have added a quantitative comparison for different clustering methods (k-means, Gaussian Mixture Model), both on the standard synthetic data showcased in Figure 3 of the submission, as well as on the CIFAR-10 dataset, using pre-trained features.
>
> The results on the standard synthetic data showcased in Figure 3 of our original submission have been added to our manuscript - see Figure 1.(a) in the attached document.
>
> The results for CIFAR-10 are as follows, they have been added to our manuscript
>
> +++++
>
> CIFAR-10 (ResNet50 for embeddings) [metric: L2 distance in coincidence matrix]:
>
> Differentiable clustering: 0.097 (ours)
>
> Classification embeddings + k-means: 0.104 (non-differentiable)
>
> Classification embeddings + GMM: 0.107 (non-differentiable)
>
> +++++
>
> This allows us to highlight that our method not only allows us to perform differentiable clustering, it allows us to learn good representations of the data (see following point)
>
> ### *Performance on more complicated datasets*
>
> We have added experiments on the CIFAR-10 dataset for both the fully supervised and semi-supervised settings (as well as a downstream classification transfer learning task, as in Figure 4 of the submission).
>
> For supervised clustering, we obtain a clustering error of 6.7% on the test set, in line with our results on MNIST and Fashion-MNIST, we also provide in Figure 1.(b) the t-SNE of embeddings obtained in this manner in the feature space.
>
> For semi-supervised clustering, the results are in Figure 2 in the attached document and have been included in the manuscript (analogous to Figure 4 on MNIST in our original submission). We observe analogous trends for clustering error to those detailed in Sections 4.2 (& 10.2) of the submission, with our model outperforming the baseline. The model attained a comparable classification error via a linear head (downstream transfer learning), to that of a baseline classification model for k=0, and for k=3, 6 outperformed the baseline.

---

### Author Rebuttal · Authors · 2023-08-09

We would like to thank the reviewers and the area chair for their overall positive evaluation of our work.

We are thankful for the high-quality, informative reviews, and their constructive suggestions. We have made the following changes, further described in individual replies to reviewers:

- We have added an additional clustering experiment on the CIFAR 10 dataset, in both the fully-supervised and semi-supervised settings. The results are in the attached document (Figure 2) and have been included in our manuscript.

- We have added a quantitative comparison for the clustering methods contained in Figure 3 of the submission, as well as evaluating different clustering methods on features of the pre-trained CIFAR-10 features. We detail these results in replies below and they have been included in our manuscript.

- We have made suggested changes: clarifying some points, and improving our terminology and presentation (see replies below for details).

---

### Decision · Program_Chairs · 2023-09-21

**Decision:**

Accept (poster)

**Comment:**

AC agreed with the reviewers that the approach is novel and interesting. However, AC remains concerned about the paper has only MNIST experiments and then in the rebuttal added cifar10. CIFAR10 is not a convincing dataset either. Ac recommends larger datasets to affirm the validity of the approach in the camera ready but recommends accepting the paper due to the reviewers' alignment to accept the paper.